# HIF1α-PHD1-FOXA1 Axis Orchestrates Hypoxic Reprogramming and Androgen Signaling Suppression in Prostate Cancer

**DOI:** 10.3390/cells14131008

**Published:** 2025-07-02

**Authors:** Limiao Liang, Dandan Dong, Jiaxue Sun, Qin Zhang, Xiayun Yang, Gong-Hong Wei, Peng Zhang

**Affiliations:** 1MOE Key Laboratory of Metabolism and Molecular Medicine, Department of Biochemistry and Molecular Biology, School of Basic Medical Sciences, Fudan University Shanghai Cancer Center, Shanghai Medical College of Fudan University, Shanghai 200032, China; 21111010013@m.fudan.edu.cn (L.L.); 19111010023@fudan.edu.cn (D.D.); 22211010021@m.fudan.edu.cn (J.S.); 2Disease Networks Research Unit, Faculty of Biochemistry and Molecular Medicine & Biocenter Oulu, University of Oulu, 90220 Oulu, Finland; qin.zhang@oulu.fi (Q.Z.); xiayun.yang@oulu.fi (X.Y.); 3State Key Laboratory of Common Mechanism Research for Major Diseases, Suzhou Institute of Systems Medicine, Chinese Academy of Medical Sciences & Peking Union Medical College, Suzhou 215123, China

**Keywords:** HIF1α, cistrome reprogramming, PHD1, FOXA1, prostate cancer

## Abstract

Hypoxia is a hallmark of aggressive prostate cancer, but how it disrupts lineage-specific transcriptional programs to drive progression remains unclear. Here, we identify the HIF1α-PHD1-FOXA1 axis as a critical mediator of hypoxic adaptation and androgen signaling suppression. Using genome-wide profiling, we demonstrate that hypoxia reprograms HIF1α chromatin occupancy, shifting its cooperation from AR to FOXA1. Mechanistically, HIF1α physically interacts with FOXA1, destabilizing it via PHD1-mediated hydroxylation—a previously unrecognized post-translational regulatory node. Under hypoxia, loss of FOXA1 attenuates androgen-responsive transcription while activating hypoxia-inducible genes, demonstrating a dual role for this axis in hypoxia adaptation and prostate cancer progression. Genetic or pharmacological disruption of HIF1α-PHD1-FOXA1 impairs prostate cancer proliferation and migration, underscoring its translational relevance. Our findings establish oxygen-dependent FOXA1 degradation as a linchpin connecting microenvironmental stress to transcriptional plasticity in advanced prostate cancer, offering new therapeutic avenues.

## 1. Introduction

Prostate cancer remains one of the most prevalent malignancies and a leading cause of cancer-related deaths among men globally, accounting for approximately 1.5 million new cases and 397,000 deaths annually [1]. Its incidence is closely associated with age, ethnicity, and geographic variation [2,3], reflecting a complex interplay of genetic predisposition and environmental factors. While landmark studies have uncovered key oncogenic events, including TMPRSS2-ERG gene fusions and alterations in PI3K signaling via PTEN deletion [4,5], the molecular mechanisms driving tumor progression and therapeutic resistance, particularly in advanced stages, remain incompletely understood.

Androgen receptor (AR) signaling is a central driver of both prostate development and prostate cancer pathogenesis [6]. Androgen Deprivation Therapy (ADT), the standard initial treatment for hormone-sensitive prostate cancer, induces tumor regression by suppressing AR activity [7]. However, disease recurrence in castration-resistant prostate cancer (CRPC) is almost inevitable in a large subset of patients. CRPC is a lethal form of the disease characterized by restored AR activity despite low circulating androgens, often through mechanisms such as AR amplification, ligand-binding domain mutations, constitutively active AR splice variants, and intratumoral androgen biosynthesis [6]. While these AR-centric mechanisms have been extensively explored, the influence of tumor microenvironmental factors, especially hypoxia, has received comparatively less attention.

Hypoxia is a hallmark of solid tumors and contributes to therapeutic resistance, metastasis, and poor prognosis [8,9]. Oxygen deficiency arises from the rapid proliferation of tumor cells outpacing vascular supply [9,10], triggering widespread alterations in gene expression and cell behavior. Hypoxia-inducible factor 1-alpha (HIF1α) is the master regulator of cellular adaptation to hypoxia [11,12]. Hypoxia-inducible factors promote epithelial-to-mesenchymal transition (EMT), invasion, and resistance to therapy in various cancers [13,14]. The regulation of HIF1α is tightly controlled by oxygen-sensing prolyl hydroxylases (PHD1–3), which target HIF1α for proteasomal degradation under normoxia [15,16]. Beyond HIF1α, oxygen-dependent modulation of other key proteins, including p53, FOXO3a, and Cep192, has also been implicated in tumorigenesis and metastasis [17,18,19,20].

Among nuclear transcription factors modulating AR signaling, FOXA1 plays a pivotal role in maintaining lineage identity and chromatin accessibility in prostate epithelial cells. Frequently mutated in prostate, breast, bladder, and salivary gland tumors [21,22], FOXA1 is known to reprogram AR-binding landscapes and influence tumor phenotypes. In breast cancer, FOXA1-driven enhancer reprogramming has been shown to activate HIF2α via super-enhancer engagement, thereby linking FOXA1 to hypoxia-associated oncogenic programs [23]. In prostate cancer, FOXA1 co-occupies reprogrammed AR-binding sites [24] and represses an intragenic enhancer within the HIF1A locus, implicating it in the modulation of hypoxic responses [25]. However, whether FOXA1 plays a broader or context-dependent role in regulating hypoxia pathways in prostate cancer remains poorly understood.

In this study, we uncover a previously unrecognized HIF1α-PHD1-FOXA1 regulatory axis that governs the adaptation of prostate cancer cells to hypoxic stress. We demonstrate that this axis integrates microenvironmental hypoxia signaling with lineage-defining transcriptional programs, revealing a novel mechanism contributing to disease progression and resistance. Our findings provide important insights into the crosstalk between tumor-intrinsic transcriptional control and extrinsic microenvironmental stress, offering potential new targets for therapeutic intervention in advanced prostate cancer.

## 2. Materials and Methods

### 2.1. ChIP-Seq Data Processing and Analysis

Raw sequencing reads were first assessed for quality using FastQC (v0.11.9), followed by adapter and quality trimming with Trimmomatic (v0.39) [26]. High-quality reads were then aligned to the human reference genome (hg19) using the BWA aligner v0.7.18-r1243-dirty [27]. HOMER (v4.1.1) was employed for peak calling and de novo motif discovery [28]. For downstream enrichment and annotation analyses, we used a combination of SAMtools (v1.16.1) [29], BEDTools (v2.31.1) [30], CEAS (v1.0.2) [31], and the ChIPseeker (v1.44.0) [32] package from Bioconductor (v3.26.2). ChIP-seq signal profiles and heatmaps were generated using deepTools (v3.5.6) [33], and signal tracks were visualized using the Integrative Genomics Viewer (IGV v2.18.4, Broad Institute, Cambridge, MA, USA) [34].

### 2.2. Cell Culture and Treatments

Human embryonic kidney cells (HEK293T) and the prostate cancer cell line DU145 were cultured in Dulbecco’s Modified Eagle Medium (DMEM; Invitrogen, Carlsbad, CA, USA), whereas LNCaP-1F5, V16A, and PC3 prostate cancer cells were cultured according to the previously described conditions [35]. In brief, these cells were maintained in RPMI-1640 medium (Thermo Fisher Scientific, Waltham, MA, USA). All media were supplemented with 10% fetal bovine serum (FBS; Gibco) and 1% penicillin–streptomycin (Gibco, Grand Island, NY, USA). Cells were maintained at 37 °C in a humidified incubator under 5% CO_2_ and either normoxic (21% O_2_) or hypoxic (1% O_2_, balanced with 5% CO_2_ and 94% N_2_) conditions using an Invivo2 400 hypoxia workstation (Ruskinn Technologies, Bridgend, UK). For chemical treatments, cells were incubated with 50 μM dimethyloxalylglycine (DMOG; Sigma-Aldrich, St. Louis, MO, USA) for 6 h to induce hypoxia-mimicking conditions, 10–30 μM MG132 (Calbiochem, San Diego, CA, USA) to inhibit proteasomal degradation, or 100 nM dihydrotestosterone (DHT; Merck, Darmstadt, Germany) to activate androgen receptor (AR) signaling. All cell lines were routinely tested and confirmed to be free of mycoplasma contamination prior to use.

### 2.3. Plasmid Constructs and Transfection

For ectopic gene expression, the human FOXA1 coding sequence was subcloned into the pcDNA3.1-V5 expression vector (Invitrogen, Carlsbad, CA, USA). The HA-tagged HIF1α expression plasmid was obtained from Addgene (plasmid #18949, Watertown, MA, USA), while additional constructs including Myc-tagged HIF1α, V5-tagged PHD1, PHD2, and PHD3, as well as shRNA expression vectors targeting relevant genes, were generously provided by Prof. Johanna Myllyharju (University of Oulu). For inducible gene silencing or overexpression, the pLKO.1-TetON-Puro-HIF1α (Addgene #118704, Watertown, MA, USA) and EZ-Tet-pLKO-Puro (Addgene #85966, Watertown, MA, USA) vectors were utilized. Transient transfections were conducted using Lipofectamine 3000 reagent (Thermo Fisher Scientific, Waltham, MA, USA) in accordance with the manufacturer’s instructions. Transfection efficiency was optimized by adjusting DNA-to-reagent ratios and confirmed by immunoblotting or fluorescence imaging where applicable.

### 2.4. Co-Immunoprecipitation (Co-IP)

Cells were lysed in ice-cold immunoprecipitation (IP) lysis buffer containing 5 mM Tris-HCl (pH 7.4), 150 mM NaCl, 1 mM EDTA, 1% NP-40, and 5% glycerol, supplemented with protease and phosphatase inhibitors (Roche, Basel, Switzerland). Lysates were incubated on ice for 30 min with intermittent vortexing, followed by centrifugation at 12,000× *g* for 10 min at 4 °C to remove debris.

For each Co-IP reaction, equal amounts of pre-cleared protein lysate were incubated with 30 μL Dynabeads Protein G (Invitrogen, Carlsbad, CA, USA) for 1 h at 4 °C to reduce nonspecific binding. The supernatants were then incubated overnight at 4 °C with 3 μL of the indicated primary antibody. Immune complexes were captured by adding an additional 30 μL of Dynabeads Protein G and incubating for 1 h at 4 °C with gentle rotation.

Following incubation, beads were washed three times with cold IP lysis buffer. Bound protein complexes were eluted by boiling the beads for 5 min in 10 μL of NuPAGE LDS Sample Buffer (Thermo Fisher Scientific, Waltham, MA, USA) mixed with 40 μL of IP lysis buffer. Eluted proteins were resolved by SDS-PAGE and analyzed via immunoblotting using the indicated antibodies.

### 2.5. Western Blot Analysis

Cells were harvested and lysed in radioimmunoprecipitation assay (RIPA) buffer containing 50 mM Tris-HCl (pH 7.4), 150 mM NaCl, 1% NP-40, 0.5% sodium deoxycholate, and 0.1% SDS, supplemented with cOmplete™ Protease Inhibitor Cocktail (Roche, Basel, Switzerland). Lysates were incubated on ice for 30 min and clarified by centrifugation at 12,000× *g* for 15 min at 4 °C. Protein concentrations were determined using a BCA protein assay (Thermo Fisher Scientific, Waltham, MA, USA).

Equal amounts of protein (50 μg per sample) were resolved by 10% SDS-PAGE and transferred onto 0.45 μm polyvinylidene difluoride (PVDF) membranes (Millipore, Burlington, MA, USA) using a Trans-Blot SD semi-dry transfer system (Bio-Rad, Hercules, CA, USA). Membranes were blocked in 5% non-fat milk prepared in TBST (Tris-buffered saline with 0.1% Tween-20) for 1 h at room temperature, followed by overnight incubation at 4 °C with primary antibodies.

The following primary antibodies were used: anti-V5 and anti-V5-HRP (Invitrogen, Carlsbad, CA, USA), anti-Myc (Cell Signaling Technology, Danvers, MA, USA), anti-HA (ImmunoWay, Plano, TX, USA), anti-Flag (Sigma-Aldrich, St. Louis, MO, USA), anti-FOXA1 (Abcam, Cambridge, UK), anti-HIF1α (Novus or Abcam, Centennial, CO, USA or Cambridge, UK), anti-AR (Santa Cruz Biotechnology, Dallas, TX, USA), anti-hydroxyproline (Abcam, Cambridge, UK), and anti-actin-HRP (Santa Cruz Biotechnology, Dallas, TX, USA). After washing with TBST, membranes were incubated with horseradish peroxidase (HRP)-conjugated anti-rabbit or anti-mouse secondary antibodies (Invitrogen, Carlsbad, CA, USA) for 1 h at room temperature.

Immunoreactive bands were visualized using SuperSignal™ West Femto Maximum Sensitivity Substrate (Thermo Fisher Scientific, Waltham, MA, USA) and imaged with a chemiluminescence detection system (ChemiDoc MP, Bio-Rad, Hercules, CA, USA).

### 2.6. Liquid Chromatography–Mass Spectrometry (LC-MS/MS)

LC-MS/MS was carried out as described in the prior studies [36]. The FOXA1-containing protein bands were excised from SDS-PAGE gels and diced into ~1 mm^3^ pieces. Gel pieces were destained with three consecutive 5 min washes in 50 mM ammonium bicarbonate containing 40% acetonitrile, followed by an additional 5 min wash with trypsin digestion buffer (40 mM ammonium bicarbonate, 9% acetonitrile). Proteins were digested by incubating gel pieces with 5 μL of sequencing-grade trypsin (20 ng/μL; Sigma-Aldrich, St. Louis, MO, USA) in digestion buffer for 20 min at room temperature. Subsequently, 15 μL of fresh digestion buffer was added, and samples were incubated overnight at 35 °C.

Peptides were extracted by transferring the supernatant to clean tubes, followed by a second extraction using 50 μL of 0.1% trifluoroacetic acid (TFA) in 30% acetonitrile. Combined peptide extracts were dried using a vacuum concentrator (SpeedVac, Waltham, MA, USA) and reconstituted in 20 μL of 0.2% TFA for subsequent mass spectrometry analysis.

Peptide samples (5–10 μL) were injected into a nanoACQUITY UPLC system (Waters, Milford, MA, USA) coupled to a Synapt G2 Q-TOF mass spectrometer (Waters). Peptides were initially trapped on a Symmetry C18 column (0.18 × 20 mm) at 5 μL/min for 5 min and then separated on a PicoFrit analytical column (0.075 × 150 mm, New Objective, Woburn, MA, USA) packed in-house with BEH C18 resin (1.7 μm; Waters, Milford, MA, USA). Chromatographic separation was performed using an 80 min linear gradient from 3% to 40% solvent B (solvent A: 0.1% formic acid in water; solvent B: 0.1% formic acid in acetonitrile) at a flow rate of 0.3 μL/min.

MS data were acquired in data-independent acquisition mode (MSE) over an *m*/*z* range of 100–1200, with alternating low-energy (0.5 s/scan) and high-energy scans using a collision energy ramp of 18–40 V. Raw data were processed using ProteinLynx Global SERVER (PLGS) v2.5 (Waters) with automated peak detection (intensity threshold: 500 counts). Database searches were performed against the SwissProt human protein database, with a false discovery rate (FDR) threshold set at 4%. Variable modifications included phosphorylation (Ser/Thr/Tyr), N-terminal acetylation, deamidation (Asn/Gln), and oxidation (Pro/Trp/Met).

### 2.7. Quantitative Reverse Transcription PCR (RT-qPCR)

Total RNA was extracted from cultured cell lines using the RNeasy Mini Kit (QIAGEN) in accordance with the manufacturer’s instructions. On-column DNase digestion was performed using RNase-Free DNase Set (QIAGEN, Venlo, The Netherlands) to eliminate contaminating genomic DNA. First-strand cDNA was synthesized from 2 μg of total RNA using the High-Capacity cDNA Reverse Transcription Kit (Applied Biosystems, Waltham, MA, USA).

Quantitative PCR was performed using SYBR Select Master Mix (Applied Biosystems, Waltham, MA, USA) on a CFX96 Real-Time PCR Detection System (Bio-Rad, Hercules, CA, USA). Relative mRNA expression levels were calculated using the ΔΔCt method and normalized to ACTB (β-actin) as the internal control. Each reaction was run in technical triplicates, and experiments were independently repeated at least three times to ensure reproducibility.

### 2.8. Chromatin Immunoprecipitation Followed by Sequencing (ChIP-Seq)

The ChIP-seq experiments were carried out as described previously [36]. Briefly, the LNCaP 1F5 cells were cross-linked with 1% formaldehyde at room temperature for 10 min to preserve protein–DNA interactions. Cross-linking was quenched by adding 125 mM glycine for 5 min. Cells were collected by centrifugation and resuspended in hypotonic lysis buffer (10 mM KCl, 20 mM Tris-HCl pH 8.0, 10% glycerol, 2 mM DTT, supplemented with protease inhibitors) and incubated at 4 °C for 50 min with gentle rotation to isolate nuclei.

Nuclei were washed twice with ice-cold PBS and subsequently lysed in SDS lysis buffer (50 mM Tris-HCl pH 8.0, 10 mM EDTA, 0.5% SDS, with protease inhibitors). Chromatin was fragmented to an average size of ~300 bp by sonication (optimized with a Bioruptor or equivalent instrument). For immunoprecipitation, 6 μg of anti-HIF1α antibody (Abcam) was pre-bound to Dynabeads Protein G (Thermo Fisher Scientific, Waltham, MA, USA), which were washed with blocking buffer (0.5% BSA in IP buffer) and incubated overnight at 4 °C.

Soluble chromatin (250–300 μg) was diluted in IP buffer (20 mM Tris-HCl pH 8.0, 2 mM EDTA, 150 mM NaCl, 1% Triton X-100, and protease inhibitors) to a final volume of 1.4 mL and incubated overnight at 4 °C with the antibody-bead complexes. Immunocomplexes were sequentially washed six times with stringent RIPA washing buffer (50 mM HEPES pH 7.5, 1 mM EDTA, 0.7% sodium deoxycholate, 1% NP-40, 0.5 M LiCl) to remove nonspecific binding.

Chromatin was eluted in DNA extraction buffer (10 mM Tris-HCl pH 8.0, 1 mM EDTA, 1% SDS), and cross-links were reversed by incubation at 65 °C overnight in the presence of Proteinase K, RNase A, and NaCl. DNA was purified using the QIAGEN MiniElute PCR Purification Kit (QIAGEN, Venlo, The Netherlands) according to the manufacturer’s instructions.

Purified ChIP DNA was used to prepare sequencing libraries with the NEBNext^®^ Ultra™ II DNA Library Prep Kit for Illumina (New England Biolabs, Ipswich, MA, USA), following the manufacturer’s protocol including end repair, adapter ligation, size selection, and PCR amplification steps. Libraries were quantified and quality-checked prior to sequencing.

### 2.9. Cell Proliferation and Viability Assays

Prostate cancer cell lines (LNCaP 1F5, PC3, DU145, and V16A) were seeded at a density of 5 × 10^3^ cells per well in 96-well plates. Cell proliferation and viability were assessed using two complementary approaches: the XTT-based Cell Proliferation Kit II (Roche, #11465015001, Basel, Switzerland) and the IncuCyte ZOOM™ live-cell imaging system (Essen BioScience, Ann Arbor, MI, USA).

For XTT assays, absorbance measurements were recorded at specified time points using a microplate reader according to the manufacturer’s instructions. For real-time kinetic analysis, phase-contrast images were captured every 2–3 h using the IncuCyte ZOOM™ system. Cell confluency was quantified using IncuCyte analysis software (IncuCyte 2021c) to evaluate growth dynamics over time.

All experiments were performed in triplicate, and data were presented as mean ± standard error of the mean (SEM). Statistical significance was determined using a two-tailed unpaired Student’s *t*-test unless otherwise indicated.

### 2.10. Wound Healing Assay

The wound healing experiments were conducted as previously described [37]. In brief, cells were seeded at a density of 5 × 10^5^ per well in 96-well ImageLock plates (Essen BioScience, Ann Arbor, MI, USA) and cultured for 24 h to reach ~90% confluency. Uniform wounds were generated using the IncuCyte^®^ WoundMaker Tool, followed by gentle washing with phosphate-buffered saline (PBS) to remove detached cells. Subsequently, 100 μL of fresh culture medium was added to each well.

Plates were placed into the IncuCyte ZOOM™ live-cell imaging system (Essen BioScience, Ann Arbor, MI, USA) and imaged at 2–3 h intervals over a 48 h period. Wound closure was quantified using IncuCyte software, which calculates wound width or relative wound density over time to assess cell migration.

Each condition was tested in 6–8 technical replicates, and data were expressed as mean ± standard error of the mean (SEM). Statistical comparisons were performed using a two-tailed, unpaired Student’s *t*-test, with *p* < 0.05 considered statistically significant.

### 2.11. Statistics Analysis

All statistical analyses were performed using GraphPad Prism version 9.0 (GraphPad Software, San Diego, CA, USA). Data are presented as mean ± standard deviation (SD) unless otherwise specified. Comparisons between two groups were conducted using a two-tailed, unpaired Student’s *t*-test. For comparisons involving three or more groups, one-way analysis of variance (ANOVA) followed by appropriate post hoc tests was employed. Statistical significance was defined as *p* < 0.05. The following significance thresholds were used: * *p* < 0.05, ** *p* < 0.01, *** *p* < 0.001, and **** *p* < 0.0001.

## 3. Results

### 3.1. Subsection Hypoxia Drives Genome-Wide Reprogramming of HIF1α Chromatin Occupancy and Transcription Factor Cooperation in Prostate Cancer

To investigate the role of HIF1α in transcriptional reprogramming under hypoxia, we used LNCaP 1F5 prostate cancer cells [38], which constitutively express high levels of HIF1α. ChIP-seq profiling was performed to map genome-wide HIF1α-binding sites under both normoxic and hypoxic conditions. Peak calling using MACS [39] and HOMER [28] algorithms identified approximately 1000 high-confidence HIF1α-binding sites under normoxia, expanding markedly to ~6000 peaks under hypoxia. Comparative genomic annotation revealed that HIF1α binding was significantly enriched at promoter regions. Under normoxia, 10% of peaks were localized within 1 kb upstream of transcription start sites (TSSs) and 2.6% in 5′ UTRs. Upon hypoxic stimulation, these proportions increased to 17.7% and 5.0%, respectively (Figure 1A), indicating extensive oxygen-dependent reprogramming of HIF1α occupancy near gene promoters.

Meta-peak analysis centered around TSSs (±3 kb) demonstrated significantly elevated HIF1α enrichment under hypoxia, indicating enhanced transcriptional engagement (Figure 1B). De novo motif analysis using HOMER confirmed robust enrichment of the canonical hypoxia response element (HRE) motif (CACGTG), validating binding specificity (Figure 1C). Co-motif scanning further revealed a dynamic switch in transcription factor partnerships: AR motifs were predominantly enriched under normoxia, whereas FOXA1 motifs emerged as the second most enriched under hypoxia (Figure 1C). These findings suggest that HIF1α forms distinct transcriptional complexes depending on oxygen availability, potentially reprogramming gene expression in cooperation with lineage-specific regulators.

Quantitative comparison of HIF1α cistrome between oxygen conditions revealed profound binding dynamics. Hypoxia induced 5780 new HIF1α peaks while 775 normoxia-enriched sites were lost; only 202 binding sites were maintained under both conditions (Figure 1D), reflecting a dramatic redistribution of HIF1α genomic occupancy in response to hypoxia. To characterize the transcriptional consequences of these shifts, we mapped promoter-associated peaks to nearby genes. Venn diagram analysis showed that 3952 genes were uniquely associated with hypoxia-specific promoter binding, whereas 420 and 442 genes were bound exclusively under normoxia or shared across conditions, respectively (Figure 1E). These data highlight an oxygen-sensitive rewiring of the HIF1α-regulated transcriptional network.

To determine the functional implications of HIF1α cistrome remodeling, we performed pathway enrichment analysis using GREAT algorithm [40]. Genes associated with hypoxia-induced binding were enriched in pathways related to cellular hypoxia responses and metabolic adaptation, indicating that HIF1α selectively drives context-specific regulatory programs (Figure 1F). Visualization of ChIP-seq signal intensities at representative loci further confirmed dynamic redistribution and stronger recruitment under hypoxia, alongside persistent occupancy at selected sites under normoxia (Appendix A).

Collectively, our findings reveal that HIF1α binding is highly responsive to oxygen levels in prostate cancer cells, exhibiting large-scale cistrome reprogramming and context-dependent transcription factor cooperation. Under normoxic conditions, HIF1α preferentially collaborates with AR, while hypoxia shifts its interactions toward FOXA1. This plasticity enables HIF1α to orchestrate distinct transcriptional programs tailored to microenvironmental oxygen availability, potentially contributing to prostate cancer progression through enhanced survival, invasiveness, and adaptation to hypoxic stress.

### 3.2. HIF1α Physically Interacts with FOXA1 in Prostate Cancer Cells

Hypoxia, a hallmark of solid tumors, is commonly associated with elevated HIF1α expression in prostate cancer tissues [41]. As a master regulator of the cellular response to oxygen deprivation, HIF1α exerts its transcriptional activity through both direct DNA binding and cooperative interactions with other transcription factors. Our genome-wide motif analysis revealed a striking enrichment of FOXA1 motifs at hypoxia-induced HIF1α-binding sites (Figure 1C), suggesting potential functional cooperation between HIF1α and FOXA1.

To determine whether this cooperation involves direct physical interaction, we first performed co-immunoprecipitation (Co-IP) assays in HEK293T cells co-transfected with Myc-tagged HIF1α and V5-tagged FOXA1. Immunoprecipitation of Myc-HIF1α followed by western blotting with anti-V5 antibody confirmed the physical association between the two proteins (Figure 2A). Reciprocal Co-IP using anti-V5 antibody also captured Myc-HIF1α (Appendix A), corroborating the interaction.

To validate these findings in a disease-relevant context, we performed endogenous Co-IP assays in LNCaP 1F5 prostate cancer cells. Cells were pre-treated with the proteasome inhibitor MG132 to stabilize HIF1α levels. Immunoprecipitation of endogenous HIF1α pulled down FOXA1 (Figure 2B), while reciprocal immunoprecipitation of FOXA1 confirmed specific co-precipitation of HIF1α (Figure 2C), demonstrating bidirectional interaction in prostate cancer cells. Notably, this interaction was markedly enhanced under hypoxic conditions (1% O_2_, 24 h), consistent with increased HIF1α protein stabilization and functional engagement (Figure 2D).

To map the domain(s) of FOXA1 responsible for this interaction, we generated three V5-tagged FOXA1 truncation mutants corresponding to its N-terminal domain (NTD), forkhead (FH) domain, and C-terminal domain (CTD), and co-expressed them with HA-tagged HIF1α in HEK293T cells [42] (Appendix A). Surprisingly, Co-IP assays revealed that HIF1α interacted exclusively with full-length FOXA1, while none of the individual domains supported detectable binding (Appendix A). This result suggests that the interaction may require the intact tertiary structure of FOXA1 or rely on additional co-factors or post-translational modifications that are absent in truncated constructs.

Collectively, these findings establish FOXA1 as a novel HIF1α-interacting partner in prostate cancer cells and suggest that their cooperation under hypoxic conditions may be structurally or contextually regulated, potentially contributing to the reprogramming of transcriptional networks in response to tumor hypoxia.

### 3.3. HIF1α Negatively Regulates FOXA1 Protein Stability and Suppresses Androgen Signaling Under Hypoxia

Hypoxia, a hallmark of solid tumors, is commonly associated with elevated HIF1α expression in prostate cancer tissues [41]. As a master regulator of the cellular response to oxygen deprivation, HIF1α exerts its transcriptional activity through both direct DNA binding and cooperative interactions with other transcription factors. Our genome-wide motif analysis revealed a striking enrichment of FOXA1 motifs at hypoxia-induced HIF1α-binding sites (Figure 1C), suggesting potential functional cooperation between HIF1α and FOXA1. Building on the physical interaction between HIF1α and FOXA1, we next examined whether hypoxia modulates FOXA1 protein stability and its downstream transcriptional consequences. Functional characterization of LNCaP 1F5 cells exposed to hypoxia (1% O_2_, 24 h) revealed a marked reduction in FOXA1 protein levels (Figure 2E), suggesting that HIF1α reduces FOXA1 protein stability. To dissect the underlying mechanism, we used the prolyl hydroxylase (PHD) inhibitor dimethyloxalylglycine (DMOG) to pharmacologically stabilize HIF1α by preventing its degradation [15], thereby mimicking hypoxia. Following 6 h of DMOG treatment, LNCaP 1F5 cells exhibited robust HIF1α stabilization alongside a pronounced reduction in FOXA1 levels (Figure 2F). This reciprocal relationship was similarly observed in two additional prostate cancer cell lines, PC3 and DU145 (Figure 2G), indicating that HIF1α-mediated suppression of FOXA1 is a broadly conserved response.

To explore how hypoxia shifts HIF1α‘s collaboration preference from AR to FOXA1 and its impact on gene expression, we evaluated changes in androgen-responsive genes and hypoxia-inducible genes. This analysis revealed decreased expression of the canonical androgen-responsive gene TMPRSS2 (Figure 2H). In contrast, mRNA levels of classical hypoxia-inducible genes, including VEGFA and BNIP3, were significantly upregulated (Figure 2I,J), suggesting that hypoxia promotes transcriptional reprogramming that dampens androgen signaling while enhancing adaptive responses to oxygen deprivation. Consistent transcriptional changes were observed, with decreased KLK3 (PSA) and increased VEGFA expression following DMOG treatment (Appendix A), reinforcing the notion that HIF1α antagonizes androgen signaling while activating hypoxia-adaptive gene programs.

Given that hypoxia is associated with prostate cancer progression [43], we compared the expression of AR-responsive genes and hypoxia-inducible genes between primary prostate cancer and CRPC patients. In the MCTP [44] and GSE32269 [45] cohorts, AR-responsive genes (TMPRSS2, KLK3, NKX3-1, and PMEPA1) were significantly downregulated in CRPC (Figure 2K–M; Appendix A). Conversely, hypoxia-inducible genes (VEGFA, PDGFB, SLC2A1, ANGPT2, EPO, FASN, MMP2, MMP9, and LDHA) showed increased expression in CRPC (Figure 2N–P; Appendix A). These clinical findings align with our cell line data, demonstrating that hypoxia and HIF1α activation promote prostate cancer progression by suppressing AR targets while activating hypoxia-inducible pathways.

To further delineate the spatial regulation of FOXA1, we performed subcellular fractionation in V16A cells cultured under normoxia or hypoxia. Hypoxia induced a striking reduction in nuclear FOXA1 protein, while nuclear HIF1α levels were substantially increased (Figure 3A). Similar results were obtained in LNCaP 1F5 cells treated with DMOG (Figure 3B), confirming the repressive effect of stabilized HIF1α on nuclear FOXA1 abundance.

To directly test whether HIF1α mediates this suppression, we employed a doxycycline-inducible lentiviral shRNA system to knock down HIF1A in LNCaP 1F5 cells (Figure 3C). HIF1α depletion led to elevated FOXA1 protein levels under normoxia. Notably, subsequent exposure to hypoxia reversed this accumulation, restoring FOXA1 suppression (Figure 3D). These data confirm that FOXA1 repression under hypoxia is mediated predominantly through HIF1α.

Altogether, our findings uncover a regulatory axis whereby HIF1α negatively regulates FOXA1 protein stability in prostate cancer cells, thereby suppressing androgen receptor signaling and promoting hypoxia-responsive transcriptional programs. This interplay between oxygen sensing and lineage-specific transcription factors highlights the HIF1α-FOXA1 axis as a critical node in the transcriptional reprogramming of prostate cancer cells under hypoxic stress.

### 3.4. PHD1 and PHD3 Selectively Regulate FOXA1 Protein Stability in Prostate Cancer Cells

Prolyl-4-hydroxylases (PHD1-3) are oxygen-sensing enzymes best known for mediating the proteasomal degradation of HIF1α through oxygen-dependent proline hydroxylation, a critical post-translational modification that governs protein stability, enzymatic activity, and protein–protein interactions [46]. Our initial observation that global PHD inhibition using DMOG led to HIF1α stabilization but paradoxically reduced FOXA1 protein levels prompted us to investigate whether individual PHD isoforms differentially regulate FOXA1 expression in prostate cancer cells.

To delineate isoform-specific functions of PHDs in FOXA1 regulation, we employed lentiviral shRNA-mediated silencing of PHD1, PHD2, and PHD3 in V16A prostate cancer cells. Quantitative PCR confirmed efficient knockdown of all three PHD isoforms (Figure 4A). Western blot analysis revealed that silencing PHD1 or PHD3 significantly decreased FOXA1 protein levels, whereas PHD2 depletion had no discernible effect (Figure 4C). Interestingly, RT-qPCR analysis showed that FOXA1 mRNA levels were reduced upon PHD2 or PHD3 knockdown but remained unchanged with PHD1 silencing (Figure 4B), indicating that PHD1 specifically modulates FOXA1 protein abundance in a post-transcriptional manner.

Furthermore, among the three isoforms, only PHD1 knockdown led to a significant reduction in KLK3 (encoding prostate-specific antigen, or PSA) mRNA levels (Figure 4B), suggesting that PHD1 exerts unique regulatory control over FOXA1-mediated transcriptional programs. These isoform-specific effects were recapitulated in LNCaP 1F5 cells, where knockout of PHD1 or PHD3 similarly diminished FOXA1 protein expression (Figure 4D). Conversely, overexpression of PHD1 in HEK293T cells, as well as ectopic expression of PHD1 or PHD3 in V16A cells, robustly enhanced FOXA1 protein levels (Figure 4E–G), further corroborating their regulatory influence.

Collectively, these findings position PHD1 as the predominant isoform responsible for stabilizing FOXA1 at the protein level, independent of transcriptional regulation. The selective effects of PHD1 depletion on both FOXA1 protein levels and KLK3 transcription underscore its unique and functionally relevant role in modulating androgen receptor signaling pathways in prostate cancer.

### 3.5. PHD1 Physically Interacts with and Hydroxylates FOXA1 in an Oxygen-Dependent Manner

Our previous data revealed that FOXA1 protein stability is regulated by PHD1 and PHD3, with PHD1 exerting a dominant post-transcriptional effect. While no direct domain-specific interaction was detected between HIF1α and FOXA1, we hypothesized that PHD enzymes might directly bind to FOXA1, mediating its hydroxylation and subsequent degradation.

To test this, we co-transfected HEK293T cells with Flag-tagged FOXA1 and V5-tagged PHD1, PHD2, or PHD3. Co-immunoprecipitation using an anti-V5 antibody followed by immunoblotting with an anti-Flag antibody revealed a specific interaction between PHD1 and FOXA1, whereas no interaction was observed with PHD2 or PHD3 (Figure 5A). These results identify PHD1 as the only member of the PHD family that physically associates with FOXA1.

PHD enzymes catalyze proline hydroxylation of specific protein substrates in an oxygen-dependent manner [15,20,47,48]. To determine whether PHD1-mediated interaction leads to FOXA1 hydroxylation, we overexpressed Flag-tagged FOXA1 in HEK293T cells and subjected the cells to normoxic or hypoxic conditions (1% O_2_, 24 h). Immunoprecipitation of FOXA1 followed by immunoblotting with a hydroxyproline-specific antibody revealed robust hydroxylation of FOXA1 under normoxia, which was significantly reduced upon hypoxic treatment (Figure 5B). These findings were validated in LNCaP 1F5 cells, where FOXA1 hydroxylation similarly decreased under hypoxia (Figure 5C), confirming the oxygen dependency of this modification in a prostate cancer context.

To pinpoint hydroxylation sites, we performed liquid chromatography–tandem mass spectrometry (LC-MS/MS) analysis on immunoprecipitated FOXA1 from LNCaP 1F5 cells cultured under normoxia and hypoxia. Multiple oxidized FOXA1 peptides were identified, but their abundance was markedly reduced in hypoxic samples (Figure 5D), further supporting the loss of hydroxylation in low-oxygen environments.

Together, these results demonstrate that PHD1 directly interacts with and hydroxylates FOXA1, establishing a novel oxygen-sensitive regulatory axis. The decrease in FOXA1 hydroxylation under hypoxia provides a mechanistic basis for its destabilization and supports a broader role for PHD1 in regulating FOXA1 function and prostate cancer transcriptional programs in response to oxygen availability.

### 3.6. HIF1α and Prolyl Hydroxylases Promote Prostate Cancer Cell Proliferation and Migration

To assess the functional consequences of HIF1α- and PHD-mediated regulation in prostate cancer, we evaluated their effects on cell proliferation and migration using LNCaP 1F5 and V16A cell lines. As expected, treatment with dihydrotestosterone (DHT) significantly enhanced proliferation in LNCaP 1F5 cells, whereas hypoxic conditions attenuated proliferation both in the absence and presence of DHT (Figure 6A), suggesting that oxygen availability modulates androgen-driven growth responses.

We next investigated the specific contribution of individual prolyl hydroxylases to cell proliferation. Lentiviral shRNA-mediated knockdown of PHD1 or PHD3 in V16A cells resulted in a significant reduction in proliferation, whereas PHD2 knockdown had no appreciable effect compared to control cells (Figure 6B). These results were consistent with the distinct effects of each isoform on FOXA1 stability. Furthermore, treatment with the pan-hydroxylase inhibitor DMOG suppressed cell proliferation and increased cell death across multiple prostate cancer cell lines (Figure 6C–F), recapitulating the phenotypes observed with PHD1 and PHD3 knockdown.

To evaluate the impact of these pathways on cell migration, we performed wound healing assays. Silencing of PHD1 or PHD3 in V16A cells significantly impaired wound closure, indicating reduced migratory capacity (Figure 6G,H). These data parallel the proliferation phenotype and further support a role for PHD1/3 in promoting aggressive cancer cell behavior.

Given prior studies implicating HIF1α in tumor cell motility [49], we examined its role in prostate cancer migration. In LNCaP 1F5 cells, DHT stimulation enhanced wound closure, whereas HIF1α knockdown markedly reduced migration under both basal and androgen-stimulated conditions (Figure 6I,J). These results suggest that HIF1α supports prostate cancer cell motility, potentially through its regulatory effects on FOXA1 and downstream transcriptional programs.

To investigate the mechanisms by which HIF1α and PHD1 regulate cellular phenotypes, we performed siRNA-mediated knockdown of HIF1α or PHD1 in LNCaP cells and analyzed epithelial–mesenchymal transition (EMT) markers. HIF1α or PHD1 knockdown increased levels of the epithelial marker CHD1 while reducing mesenchymal markers (CHD2, VIM) and EMT-associated transcription factors (SNAIL1/2, ZEB1/2, and TWIST1) (Figure 6K,N). Furthermore, clinical analyses across multiple prostate cancer cohorts (Broad_NG2012 [50], CPGEA [22], DKFZ [51], FHCRC [52], GSE62872 [53], MCTP [44], MSKCC [54], NPC [55], SMMU [56], and TCGA [57]) revealed significant positive correlations between HIF1α/PHD1 expression and EMT scores [58], cell cycle progression (CCP) scores [59], mesenchymal markers, and EMT-associated transcription factors. Conversely, negative correlations were observed with epithelial markers (Figure 6L,M,O,P and Appendix A). These findings demonstrate that HIF1α and PHD1 promote prostate cancer progression through EMT pathway regulation.

Taken together, these findings establish HIF1α, PHD1, and PHD3 as positive regulators of prostate cancer cell proliferation and migration. The coordinated control of FOXA1 stability by these factors links oxygen sensing to androgen signaling and highlights a regulatory axis that may contribute to tumor progression and metastatic potential.

## 4. Discussion

Hypoxia is an intrinsic feature of the tumor microenvironment and plays a pivotal role in shaping the aggressive phenotype of solid tumors, including prostate cancer. Despite longstanding recognition of HIF1α as a master regulator of hypoxic adaptation, the precise mechanisms through which HIF1α influences lineage-specific transcriptional networks and therapeutic resistance in prostate cancer remain incompletely defined. Here, we uncover a dynamic and previously unappreciated HIF1α-PHD1-FOXA1 regulatory axis that orchestrates transcriptional reprogramming and cellular adaptation in response to hypoxic stress.

Our study demonstrates that hypoxia drives a dramatic redistribution of the HIF1α cistrome, with a distinct shift in co-factor preference from AR under normoxia to FOXA1 under hypoxia. This context-dependent transcriptional partnership reflects the remarkable plasticity of HIF1α in prostate cancer and adds a new dimension to our understanding of how hypoxia remodels lineage transcriptional programs. FOXA1, a pioneer factor that maintains epithelial identity and mediates AR cistrome engagement, has been previously implicated in prostate tumorigenesis and therapeutic resistance, particularly through mutations and enhancer reprogramming [21,60,61]. However, our findings reveal that FOXA1 is not merely a passive co-factor but is actively modulated by the hypoxic microenvironment.

Mechanistically, we identify PHD1 as a critical post-translational stabilizer of FOXA1 through prolyl hydroxylation, a modification that is diminished under hypoxia. This loss of hydroxylation leads to FOXA1 destabilization, providing a novel mechanism by which hypoxia represses FOXA1 protein levels. Importantly, we show that HIF1α physically interacts with FOXA1 in a hypoxia-enhanced manner and functionally suppresses FOXA1-driven transcriptional programs, including AR signaling. This is particularly striking in the context of castration-resistant prostate cancer (CRPC), where AR activity is typically reactivated through diverse mechanisms [4,6,7]. Our data suggest that hypoxia, via HIF1α, can bypass classical AR signaling by destabilizing FOXA1 and redirecting transcriptional output toward hypoxia-adaptive and pro-survival programs.

In dissecting the PHD family, we uncover isoform-specific regulation of FOXA1, with PHD1 exerting dominant control through direct interaction and hydroxylation, while PHD3 appears to regulate FOXA1 at the transcript level. These findings broaden the role of PHDs beyond their canonical function in HIF1α degradation and position PHD1 as a molecular gatekeeper of FOXA1 stability and androgen responsiveness in prostate cancer cells. The dual involvement of PHD1 in stabilizing FOXA1 and indirectly modulating HIF1α underscores its role as a central integrator of oxygen sensing and lineage control.

Functionally, both HIF1α and PHD1 promote prostate cancer cell proliferation and migration. Genetic depletion or pharmacological inhibition of HIF1α and PHD1/3 disrupts FOXA1 stability and impairs tumor cell growth and motility, highlighting the therapeutic potential of targeting this axis. These findings align with emerging evidence that metabolic and epigenetic remodeling under hypoxia contributes to lineage plasticity, immune evasion, and treatment resistance [8,11,62,63,64].

Together, our study establishes a novel oxygen-sensitive transcriptional switch in prostate cancer (Figure 7). We propose that hypoxia triggers a dual mechanism: (1) stabilization and reprogramming of HIF1α binding toward FOXA1-enriched genomic regions, and (2) destabilization of FOXA1 through impaired PHD1-mediated hydroxylation. This integrated model of cistrome reprogramming and protein turnover reveals a key molecular circuitry by which prostate cancer cells escape AR dependency and adapt to hostile microenvironments. By uncovering the HIF1α-PHD1-FOXA1 axis, our work opens new avenues for therapeutic intervention targeting transcriptional plasticity and hypoxia-driven progression in advanced prostate cancer.

## Figures and Tables

**Figure 1 cells-14-01008-f001:**
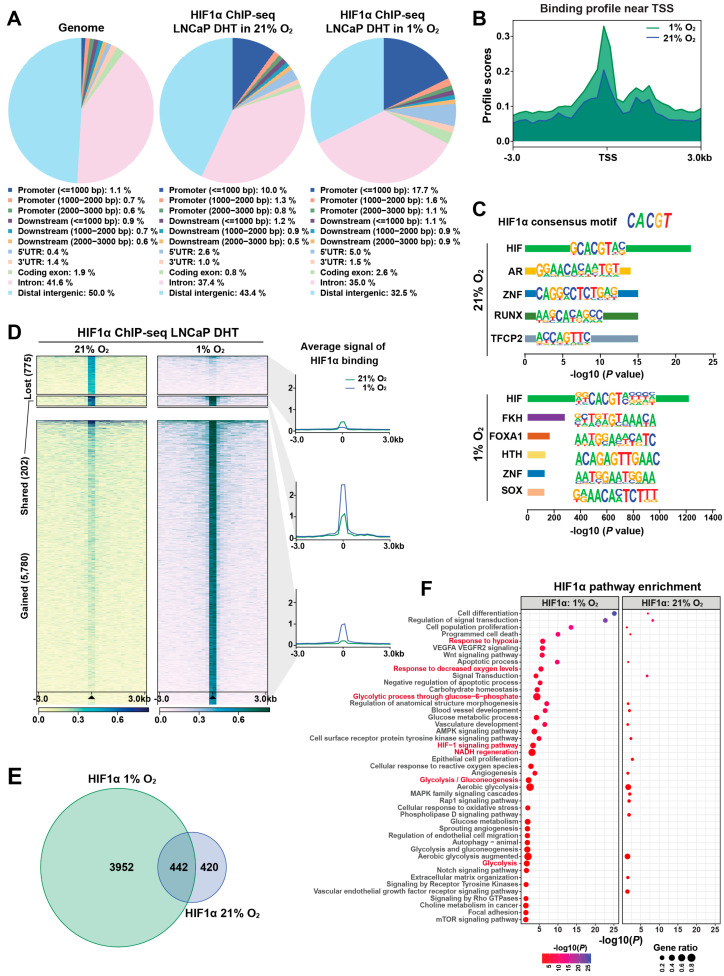
Genome-wide profiling reveals dynamic, condition-specific HIF1α cistrome and gene regulatory programs in prostate cancer cells. (**A**) Genomic distribution of HIF1α-binding sites identified by ChIP-seq in LNCaP 1F5 cells cultured under normoxic or hypoxic conditions in the presence of DHT stimulation. Peaks were annotated relative to genomic features, including promoters (defined as ±1 kb, ±2 kb, or ±3 kb from transcription start sites [TSS]), downstream elements (±1–3 kb from transcription end sites), gene bodies (including 5′UTR, 3′UTR, exons, and introns), and distal intergenic regions. The whole genome served as background control. (**B**) Aggregate binding profiles of HIF1α centered on TSS, highlighting peak enrichment and positional preference under normoxic and hypoxic conditions. (**C**) Sequence analysis of HIF1α-binding regions using HOMER identifies the canonical hypoxia response element (HRE) as the most enriched DNA-binding motif, alongside additional significantly enriched transcription factor motifs, suggesting cooperative regulatory inputs. (**D**) ChIP-seq signal intensity plots demonstrate oxygen-dependent remodeling of HIF1α occupancy at promoter regions, with hypoxia inducing substantial reprogramming of promoter-proximal binding. (**E**) Venn diagram illustrating the overlap and divergence of HIF1α-bound promoter targets between normoxic and hypoxic conditions, revealing a substantial subset of condition-specific target genes. (**F**) Functional enrichment analysis of hypoxia-specific HIF1α target genes using the Genomic Regions Enrichment of Annotations Tool (GREAT) identifies significant associations with hypoxia-responsive signaling pathways, metabolic adaptation, and stress response programs.

**Figure 2 cells-14-01008-f002:**
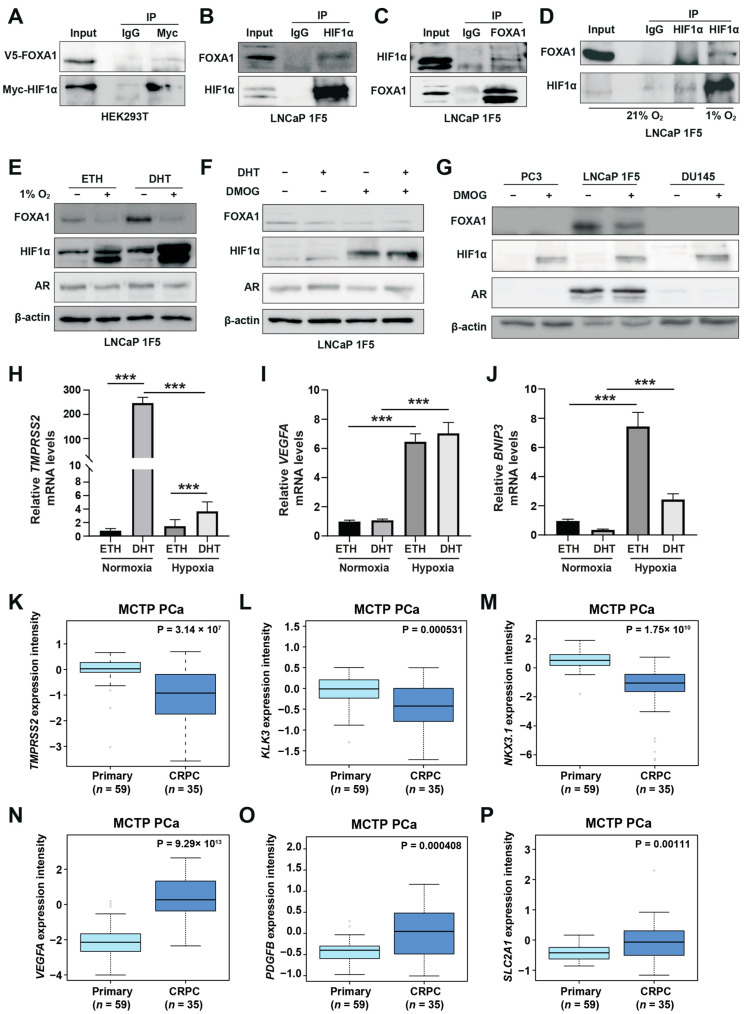
HIF1α physically interacts with and destabilizes FOXA1 in response to hypoxic stress in prostate cancer cells. (**A**) Co-immunoprecipitation (Co-IP) assay in HEK293T cells transiently co-transfected with Myc-tagged HIF1α and V5-tagged FOXA1 constructs for 48 h. Protein complexes were immunoprecipitated using an anti-Myc antibody, and both input and co-IP fractions were subjected to immunoblotting with anti-V5 and anti-Myc antibodies to confirm physical interaction between HIF1α and FOXA1. (**B**,**C**) Endogenous interactions between HIF1α and FOXA1 in LNCaP 1F5 cells were validated by reciprocal Co-IP using anti-HIF1α (**B**) and anti-FOXA1 (**C**) antibodies, followed by western blot analysis. (**D**) Hypoxia enhances the HIF1α–FOXA1 interaction in LNCaP 1F5 cells. Cells were cultured under normoxic or hypoxic (1% O_2_) conditions for 24 h, followed by co-IP using anti-HIF1α antibody and immunoblotting with anti-FOXA1 and anti-HIF1α antibodies. (**E**) Western blot analysis of LNCaP 1F5 cells treated with or without 100 nM dihydrotestosterone (DHT) under hypoxic (1% O_2_) conditions for 24 h. Protein levels of FOXA1, HIF1α, and other indicated markers were assessed to examine hormone–hypoxia crosstalk. (**F**,**G**) Western blot analysis of FOXA1 and HIF1α protein expression in LNCaP 1F5 cells, (**F**) as well as castration-resistant prostate cancer cell lines PC3 and DU145 (**G**) treated with DHT (100 nM, 24 h) and/or the HIF stabilizer DMOG (50 μM, 6 h), demonstrating modulation of FOXA1 stability in response to HIF1α activation across prostate cancer models. (**H**–**J**) Quantitative RT-PCR analysis of TMPRSS2 (**H**) and canonical HIF1α target genes VEGFA and BNIP3 (**I**,**J**) in LNCaP 1F5 cells treated with or without DHT (100 nM) and hypoxia (1% O_2_) for 24 h. Results represent mean ± SD of three independent experiments. *** *p* < 0.001 by Student’s *t*-test. (**K**–**M**) Expression profiling of AR transcriptional targets between primary prostate cancer and CRPC, including TMPRSS2 (**K**), KLK3 (**L**), and NKX3.1 (**M**). (**N**–**P**) Comparative analysis of HIF1α transcriptional targets in primary prostate cancer and CRPC.

**Figure 3 cells-14-01008-f003:**
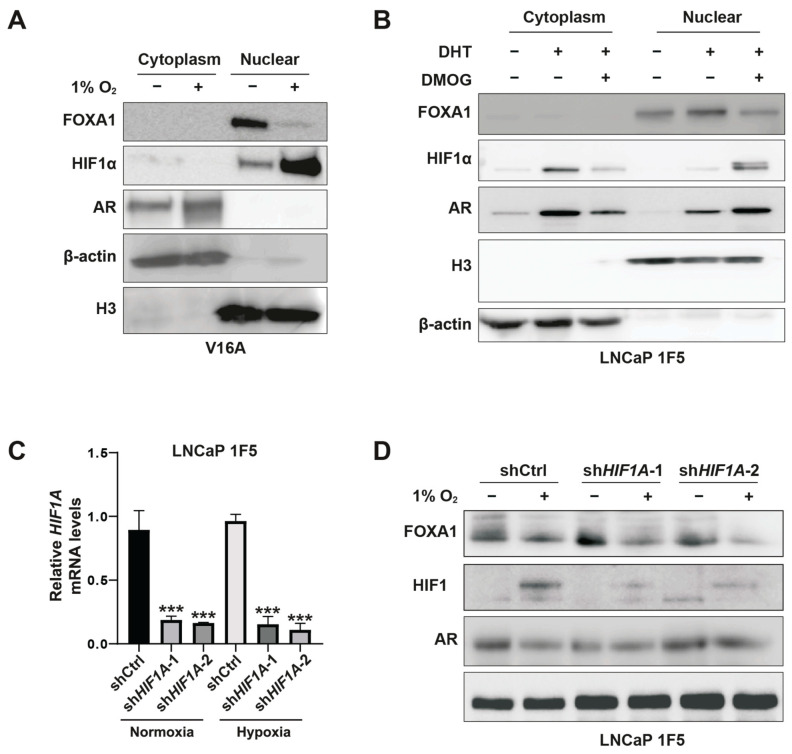
HIF1α mediates hypoxia-induced suppression of FOXA1 protein in prostate cancer cells. (**A**) V16A cells were exposed to normoxic (21% O_2_) or hypoxic (1% O_2_) conditions for 24 h. Cytoplasmic and nuclear fractions were isolated and analyzed by western blotting to assess subcellular localization and expression levels of FOXA1 and related proteins in response to hypoxia. (**B**) LNCaP 1F5 cells were treated with the HIF prolyl hydroxylase inhibitor DMOG (50 μM) for 6 h to stabilize HIF1α under normoxia. Cytoplasmic and nuclear extracts were analyzed by western blotting, revealing decreased FOXA1 protein levels upon HIF1α stabilization. (**C**) Quantitative RT-PCR analysis of HIF1A mRNA expression in LNCaP 1F5 cells transduced with control or HIF1A-targeting shRNA. Cells were cultured under normoxia or hypoxia (1% O_2_, 24 h). Data represent mean ± SD from three independent experiments. *** *p* < 0.001, Student’s *t*-test. (**D**) Knockdown of HIF1α in LNCaP 1F5 cells using lentiviral shRNA abrogates hypoxia-induced FOXA1 protein suppression. Cells were treated under normoxic or hypoxic conditions (1% O_2_, 24 h), and protein expression of FOXA1, HIF1α, and AR was assessed by western blotting.

**Figure 4 cells-14-01008-f004:**
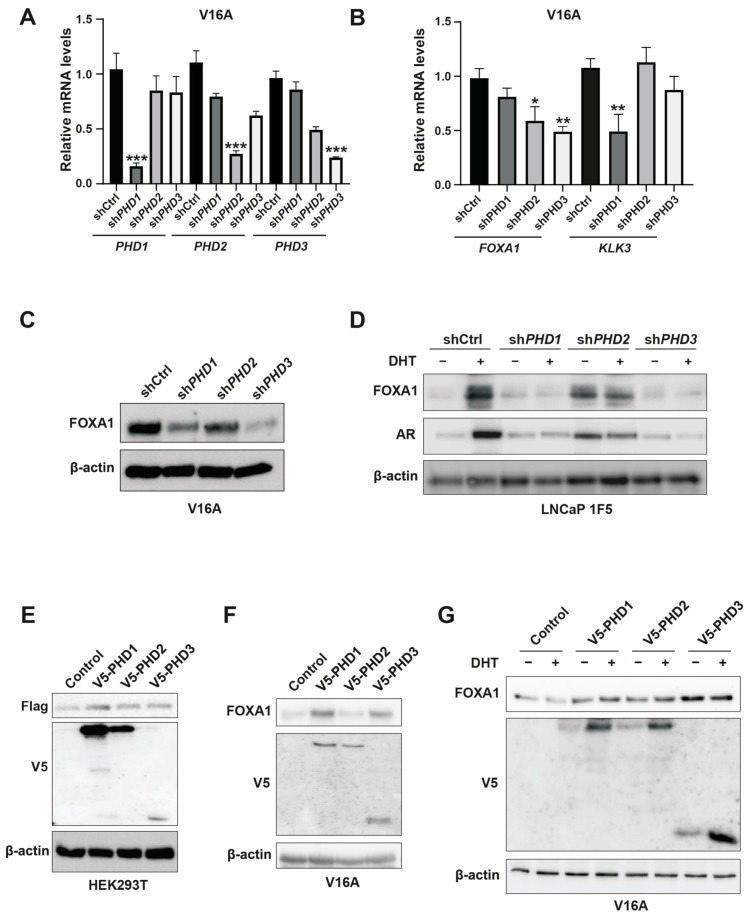
Inhibition of PHD1 and PHD3 regulates FOXA1 protein stability in prostate cancer cells. (**A**) Quantitative RT-PCR analysis confirming efficient knockdown of prolyl hydroxylase domain-containing proteins PHD1, PHD2, and PHD3 in prostate cancer cells. Data are shown as mean ± SD from three independent experiments. *** *p* < 0.001 by Student’s *t*-test. (**B**) Expression levels of KLK3 and FOXA1 mRNA following individual knockdown of PHD1, PHD2, and PHD3, as assessed by RT-qPCR. Results represent three biological replicates. * *p* < 0.05, ** *p* < 0.01. (**C**,**D**) Western blot analysis of FOXA1 protein levels in V16A cells (**C**) and LNCaP 1F5 cells (**D**) following PHD1-3 knockdown. PHD1 and PHD3 depletion led to marked changes in FOXA1 protein abundance, supporting post-transcriptional regulation. (**E**) Co-expression of Flag-tagged FOXA1 with V5-tagged PHD1, PHD2, or PHD3 in HEK293T cells. Whole-cell lysates were subjected to immunoblotting with anti-Flag and anti-V5 antibodies to evaluate potential FOXA1-PHD interactions and effects on FOXA1 stability. (**F**,**G**) V16A cells overexpressing V5-tagged PHD1, PHD2, or PHD3 with or without DHT treatment (100 nM, 24 h) were analyzed by western blotting for FOXA1 expression. Overexpression of PHD1 and PHD3 modulated FOXA1 protein levels in a context-dependent manner.

**Figure 5 cells-14-01008-f005:**
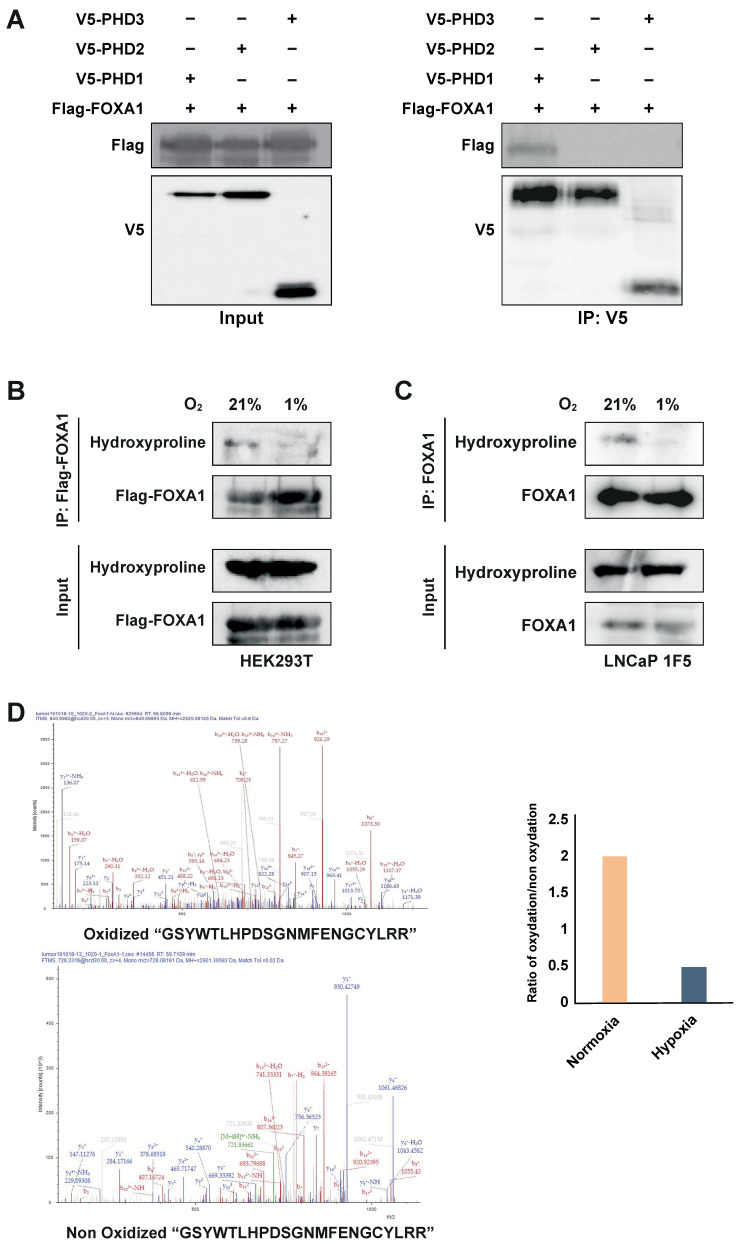
PHD1 directly interacts with and hydroxylates FOXA1. (**A**) HEK293T cells were co-transfected with Flag-tagged FOXA1 and V5-tagged PHD isoforms (PHD1, PHD2, and PHD3). Protein interactions were assessed via co-immunoprecipitation using an anti-V5 antibody, followed by immunoblotting with an anti-Flag antibody to detect FOXA1 binding. (**B**) Hydroxylation status of FOXA1 was analyzed by immunoprecipitating Flag-tagged FOXA1 from HEK293T cells cultured under normoxic (21% O_2_) or hypoxic (1% O_2_, 24 h) conditions, followed by immunoblotting with an anti-hydroxyproline antibody. (**C**) Endogenous FOXA1 was immunoprecipitated from LNCaP 1F5 cells under normoxia or hypoxia (1% O_2_, 24 h), and hydroxylation was evaluated via hydroxyproline-specific immunoblotting. (**D**) Mass spectrometry analysis of FOXA1 purified from normoxic and hypoxic cells identified hydroxylation at Proline 248 within the peptide sequence GSYWTLHPDSGNMFENGCYLRR. b4 ion fragmentation patterns (*m*/*z* values indicated) confirmed the modification site at Pro248.

**Figure 6 cells-14-01008-f006:**
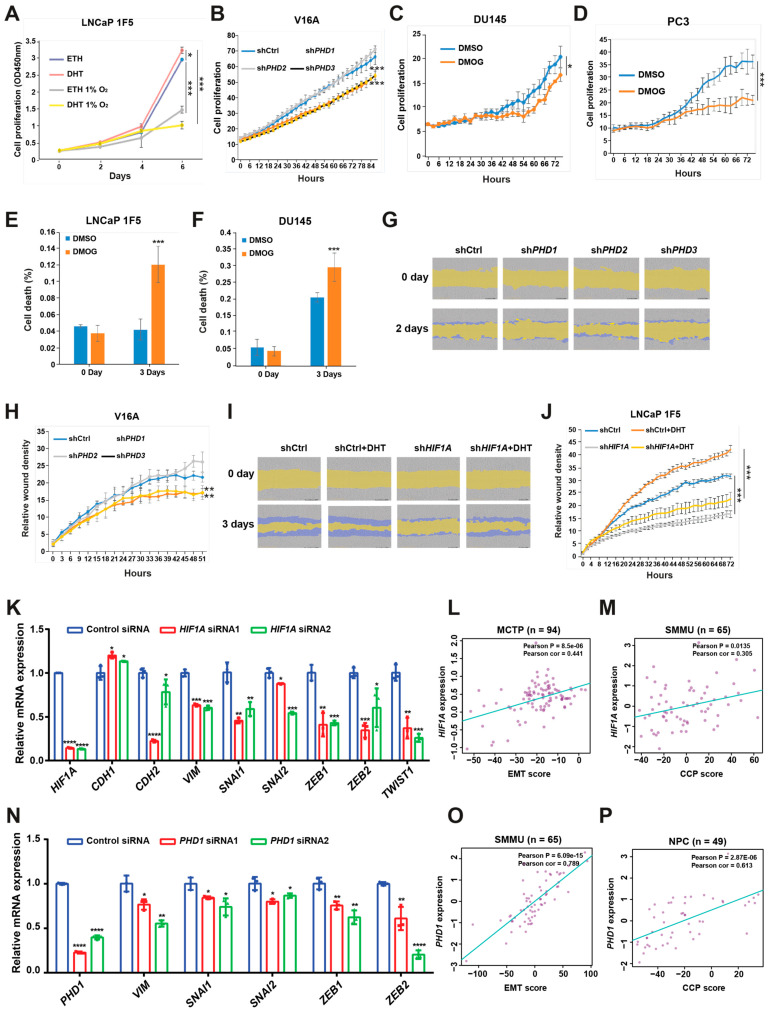
HIF1α and prolyl hydroxylases modulate prostate cancer cell proliferation and migration. (**A**) LNCaP 1F5 cells were cultured under normoxic or hypoxic (1% O_2_) conditions with or without dihydrotestosterone (DHT), and cell proliferation was quantified using the XTT assay by measuring absorbance at 450 nm. (**B**) V16A cells with stable knockdown of PHD1, PHD2, or PHD3 were monitored using the IncuCyte™ Zoom live-cell imaging system. Proliferation dynamics were assessed by quantifying percent confluence every 3 h. (**C**,**D**) Proliferation kinetics of DU145 (**C**) and PC3 (**D**) cells treated with the pan-prolyl hydroxylase inhibitor DMOG (50 μM) were measured via live-cell imaging (IncuCyte™), with confluence recorded at 3 h intervals over a 72 h period. (**E–F**) Viability analysis of LNCaP 1F5 (**E**) and DU145 (**F**) cells treated with DMOG (50 μM), visualized by phase-contrast and fluorescent microscopy using a live/dead viability dye. Quantification of dead cells was performed in DU145 and PC3 cells after 72 h of treatment. (**G**–**J**) Cell migration assays were conducted in PHD1-, PHD2-, and PHD3-knockdown V16A cells (**G**,**H**) and HIF1A-knockdown LNCaP 1F5 cells (**I**,**J**) using the IncuCyte™ platform to evaluate real-time motility. Data are representative of at least three independent biological replicates. (**K**) RT-qPCR analysis was performed to elucidate the relationship between the expression of HIF1A and genes associated with EMT. (**L**,**M**) Correlation of HIF1A expression with EMT score (**L**) or cell cycle progression (CCP) score (**M**) across distinct prostate cancer cohorts. (**N**) mRNA expression analysis of EMT-associated genes following PHD1 knockdown. (**O**,**P**) Correlation of PHD1 expression with EMT score (**O**) or cell cycle progression (CCP) score (**P**) across distinct prostate cancer cohorts. Statistical significance was assessed using appropriate tests, with *p*-values indicated as * *p* < 0.05, ** *p* < 0.01, *** *p* < 0.001, **** *p* < 0.0001.

**Figure 7 cells-14-01008-f007:**
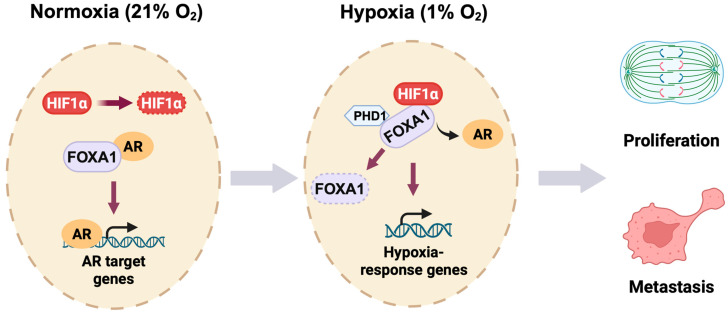
Oxygen-responsive reprogramming of transcription in prostate cancer. Under normoxic conditions (21% O_2_), HIF1α undergoes rapid PHD-mediated hydroxylation and subsequent proteasomal degradation in prostate cancer cells. FOXA1 interacts with the androgen receptor (AR) to promote AR target gene expression. During hypoxia (1% O_2_), stabilized HIF1α binds FOXA1 and inhibits PHD1-dependent hydroxylation of FOXA1, leading to FOXA1 degradation. This HIF1α-PHD1-FOXA1 axis facilitates a transcriptional switch from AR-dependent signaling to hypoxia-responsive gene activation, ultimately promoting prostate cancer cell proliferation and metastatic progression.

## Data Availability

The datasets used and analyzed in this paper are available from the corresponding author upon reasonable request.

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
