# Peer review of "HIF1α-PHD1-FOXA1 Axis Orchestrates Hypoxic Reprogramming and Androgen Signaling Suppression in Prostate Cancer"

_cells, 2025, doi:10.3390/cells14131008_

Round 1
Reviewer 1 Report
Comments and Suggestions for Authors
The manuscript, “HIF1 apha-PHD1-FOXA1 axis orchestrates hypoxic reprogramming and androgen signaling in prostate cancer” by Liang et al., report hypoxia driven androgen signaling suppression resulting from HIF1 alpha/PHD1 mediated destabilization of FOXA1. The study objectives have been addressed using a variety cell culture models and relevant experiments.
Overall research design and data interpretation are good.
Following questions and clarifications need authors’ response:
- Figure 2D FOXA1 panel picture is not clear and needs replacement.
- A schematic diagram of a model derived from this study will enhance the manuscript.
- Although authors talk about translational value of the study, they did not perform such study. They can easily address this by in silico analyses of AR and HIF1 alpha transcriptional targets (TMPRSS2, PSA, PMEPA1, VEGF, BNIP3 etc) between primary prostate cancers and CRPC.
Reviewer 2 Report
Comments and Suggestions for Authors
In the present study, the authors provided evidences that HIF1α could interact with FOXA1 and destabilized it via PHD1-mediated hydroxylation. The protein-protein interaction was demonstrated by co-IP in HEK293 and LNCaP. The experimental designs for molecular mechanism identification are well done. However, this manuscript contains flaws requiring to be revision.
Major concerns
1. I observed that some HIFα western blot showed multiple bands but it showed single band in other results. Could author explain it?
2. The authors mention that "loss of FOXA1 under hypoxia attenuates androgen-responsive transcription while activating hypoxia-inducible genes, revealing a dual role for this axis in metabolic adaptation and therapeutic resistance." But, I don't find any functional assay associated with metabolic adaptation and therapeutic resistance. The authors just examine the effect of knockdown PDH or HIF1α on proliferation and migration. The results is inconsistent with abstract. It would better to further examine the expression of EMT markers or AR-response genes expression in knockdown PDH and HIF1α experiments.
Reviewer 3 Report
Comments and Suggestions for Authors
In this manuscript, the authors describe the interplay between HIF1α, PHD1 and FOXA1 in prostate cancer under hypoxic conditions. The results presented are clear and comprehensive. The authors applied innovative and suitable methods for the purpose of this research, such as ChIP-seq data processing and analysis, transfection of prostate cancer cell lines at different degrees of differentiation, co-immunoprecipitation (Co-IP), liquid chromatography-mass spectrometry (LC-MS/MS), quantitative reverse transcription PCR (RT-qPCR) and cellular assays. The results obtained describe a new molecular mechanism of prostate cancer (PCa) that is resistant to hormones and occurs in a hypoxic environment, which is typical of the most aggressive forms of the disease. This also made it possible to identify new potential pharmacological targets.
Round 2
Reviewer 2 Report
Comments and Suggestions for Authors
The authors had provided appropriate responses to all questions. I agree that this manuscript meets the criteria for publication.